# META-REWARDING LANGUAGE MODELS:
## *Self-Improving Alignment with LLM-as-a-Meta-Judge*

## ABSTRACT

Large Language Models (LLMs) are rapidly surpassing human knowledge in many domains. While improving these models traditionally relies on costly human data, recent self-rewarding mechanisms (Yuan et al., 2024c) have shown that LLMs can improve by judging their own responses instead of relying on human labelers. However, existing methods have primarily focused on improving model responses rather than judgment capabilities, resulting in rapid saturation during iterative training. To address this issue, we introduce a novel *Meta-Rewarding* step to the self-improvement process, where the model judges its own judgements and uses that feedback to refine its judgment skills. Surprisingly, this unsupervised approach improves the model's ability to judge *and* follow instructions, as demonstrated by a win rate improvement of Llama-3-8B-Instruct from 22.9% to **39.4%** on AlpacaEval 2, and 20.6% to **29.1%** on Arena-Hard. These results strongly suggest the potential for self-improving models without human supervision.

## 1 INTRODUCTION

Large Language Models (LLMs) are advancing significantly in their ability to follow instructions and respond to user queries (OpenAI, 2023; Touvron et al., 2023). An important phase in training these models is instruction tuning (Ouyang et al., 2022), which typically involves training LLMs on datasets curated by humans, either via supervised finetuning or preference optimization. Nevertheless, the acquisition of human-generated data is both costly and time-consuming. Furthermore, the quality of such data is inherently constrained by the limitations of human capabilities. The so-called 'Super Alignment' challenge (Burns et al., 2023) aims to find a solution to steering or controlling potentially super-intelligent AIs when their actions are inherently beyond human abilities to judge.

Among the potential solutions to this challenge, self-judging by the AI emerges as a particularly promising approach. Yuan et al. (2024c) introduces an iterative *Self-Rewarding* mechanism that enables an LLM to improve autonomously. The process involves a single model that takes on two distinct roles, as an actor and as a judge. As an *actor*, the model produces responses that are aimed to fulfill specific instructions. As a *judge* (a special kind of acting), the model evaluates these responses via LLM-as-a-Judge prompting (Zheng et al., 2024) and assigns rewards. The objective of the actor during this self-play is to maximize its reward, thereby improving its ability to follow instructions.

We hypothesize that a major limitation of this previous work is that its learning objective enhances the model's ability as an actor to generate better responses, while overlooking improving the model's ability as a judge. If the ability to judge does not improve then training the actor over iterations can quickly saturate – or worse could overfit the reward signal, *a.k.a.* reward hacking. Consequently, it is imperative to also improve the model's capabilities as a judge in addition to its ability to act.

In this paper, we propose a novel method called *Meta-Rewarding* which assigns rewards to its own judgements to train the model's ability to judge. The key idea is to introduce a third role of *meta-judge*, whose task is to evaluate the model's own judgements. While the judge evaluates the actor's responses, the meta-judge evaluates the judge's judgments (including rewards that it assigns) using a mechanism similar to LLM-as-a-Judge, which we term *LLM-as-a-Meta-Judge*. The meta-judge enables us to build training data containing preference pairs of judgements, in addition to the standard preferences between actor responses derived from the standard judge. Our Meta-Rewarding method thus aims to explicitly improve both the acting and judging skills of a model – whereby these combined skills should help to enhance its instruction following ability as an actor. It is important to

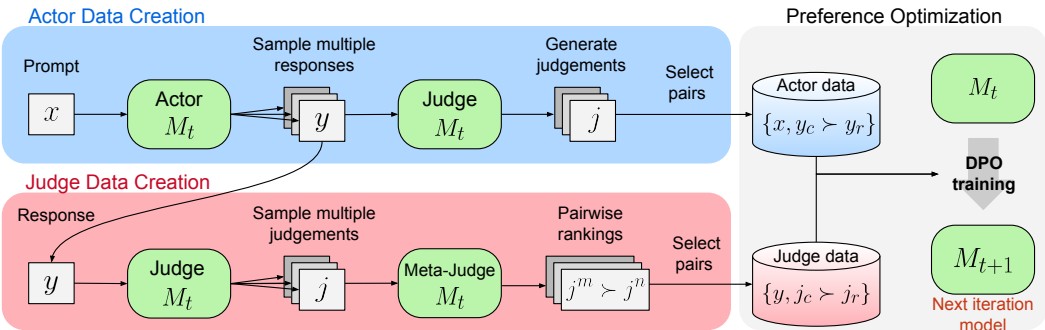

Figure 1: **Meta-Rewarding iterative training scheme.** The language model at step $t$ behaves as an *actor* to generate responses to instructions, as a *judge* to assign rewards to those responses, and as a *meta-judge* to evaluate its own judgments. The judgments are used to create preference pairs to improve its ability to act, and the meta-judgments are used to create preference pairs to improve its ability to judge. Both preference pair sets are used together to train the model for the next iteration.

note that all three roles - *actor*, *judge*, and *meta-judge* - are performed by the same model, thereby maintaining a self-improving nature that requires no extra human data.

In addition to enhancing the judging ability through Meta-Rewarding, we also address the length-bias issue in the judging process (Singhal et al., 2023). Like other reward models, the judge tends to favor long responses, which can make response length grow during iterative DPO (Yuan et al., 2024c). To counteract this, we combine the judge score with length information to determine the winning response, ensuring that a shorter response is chosen when scores are close.

In our experiments we start from Llama-3-8B-Instruct and perform multiple iterations of our Meta-Rewarding training. When evaluated on AlpacaEval 2 (Dubois et al., 2024b), we see a substantial improvement in the length-controlled (LC) win rate (from 22.9% to 39.4%), even outperforming GPT-4-0314[1]. We also observe that our method outperforms standard Self-Rewarding training even if it is enhanced with our length-bias improvements (35.5% vs 39.4%), highlighting the importance of the meta-judge. We also see similar improvement on Arena-Hard benchmark (Li et al., 2024), which is a benchmark targeting models' ability to answer complex and hard questions.

## 2 META-REWARDING

In our method, we assume a setup where we only have an initial seed model, an instruction-tuned LLM, and no further human supervised training data. The idea is to generate training data from the model itself through an iterative self-play process. In this process, the model assumes three main roles: as an actor, it generates responses to given prompts; as a judge, it evaluates and scores its own responses; and as a meta-judge, it compares the quality of its own judgments.

While training the actor to generate better responses to user queries is the final objective, this training's efficacy relies on the accuracy of the judge. As the judge's accuracy increases, it will provide higher quality feedback for training the actor, ultimately leading to a better actor. Therefore, the goal of Meta-Rewarding is to improve the model's capability both as actor and judge during training. The role of the meta-judge is to provide feedback necessary for training the judge.

At a high level, as depicted in Figure 1, our method is an iterative training scheme that starts from a given seed LLM, which assumes all three roles. An iteration starts with the actor generating multiple response variations for each prompt. This is followed by the judge evaluating each response using an LLM-as-a-Judge prompt and generating a judgement that contains a score. This score then allows us to build preference pairs of responses for training the actor. For training the judge, we pick a single response and let the meta-judge compare two of its judgement variations generated by the judge to determine which one is better using an LLM-as-a-Meta-Judge prompt, see Figure 2. This step enables us to create preference pairs of judgements that can be used for training the judge.

---

[1]https://tatsu-lab.github.io/alpaca_eval/

---

**LLM-as-a-Meta-Judge Prompt**

{Meta-Judge Instruction Rubric}

User: {prompt}
Response: {response}

Judgment A: {judgment_a}
Judgment B: {judgment_b}

{Additional Instructions}

{Force output format}: "Winner: [Judgement A | Judgement B]"

---

Figure 2: Prompt used by the meta-judge to compare given two judgements (Section A.2).

Once we have the preference data both for the actor and the judge, then we apply preference optimization on the dataset via DPO (Rafailov et al., 2024)[2]. After the training, we end up with an improved model that will be then used for the next iteration, both for generating training data and as an initial model for the optimization. Next, we will describe each preference data creation process in detail.

## 2.1 ACTOR PREFERENCE DATASET CREATION

Our approach to create the actor preference dataset on a given iteration is built upon the pipeline introduced by Yuan et al. (2024c), with a crucial modification to incorporate a length-control mechanism. Section 3.5 proves this change to be essential in preventing the responses from lengthening and improving the length-controlled win rate. The dataset creation process consists of three steps:

**Sample Responses from Actor.** We assume we have a given set of prompts. For each prompt $x$, we generate $K$ different responses $\{y_1, \ldots, y_K\}$ by sampling from the current model $M_t$ at iteration $t$.

**Aggregate Multiple Judgments.** For each response $y_k$, we generate $N$ different judgments $\{j_k^1, \ldots j_k^N\}$ from $M_t$ using an LLM-as-a-Judge prompt (shown in Section A.1). The prompt instructs the model to evaluate the given response $y_k$ for prompt $x$ according to a fixed rubric and output its chain-of-thought reasoning and a final score out of 5. We use regular expressions to parse the scores, discarding any judgments with parsing errors or those not adhering to the 5-point scale. The final reward score for each response is then calculated by averaging all valid judgment scores.

**Preference Data Selection with Length-Control.** The previous work simply selects the highest $S_{\max}$ and lowest $S_{\min}$ scored responses as the chosen $y_c$ and rejected $y_r$ as a preference pair for each prompt. However, this leads to length explosion where responses get longer with each iteration, due to the length-bias of the judge (Dubois et al., 2024a; Park et al., 2024; Yuan et al., 2024b). To mitigate this, we introduce a simple length-control mechanism. We define a quality tier parameter $\rho \in [0, 1]$ to control the trade-off between score-based selection and length consideration. Responses with scores in the top tier, specifically within the range $[(1 - \rho)S_{\max} + \rho S_{\min}, S_{\max}]$, are considered to have similar quality. For selecting the chosen response $y_c$, we opt for the shortest response within this top tier. This approach helps to counteract the tendency of judges to favor longer responses, which can lead to biased training data. Conversely, for the rejected response $y_r$, we select the longest response with a score in the range $[S_{\min}, (1 - \rho)S_{\min} + \rho S_{\max}]$. Setting $\rho$ to 0 effectively disables the length-control, reverting to a purely score-based selection.

## 2.2 JUDGE PREFERENCE DATASET CREATION

Unlike the judge that provides score-based judgements, we design the meta-judge to operate in a pairwise mode by comparing two given judgements.

**Response Selection:** To prepare effective training data for the judge, we focus on responses where the judge is the least certain, as measured by the variance of the scores it has given. To be more specific, we first compute the score variance given by the $N$ different judgments for every response

---

[2]Note that while other RLHF methods can be employed, we chose to use DPO because of its simplicity and stability.

$y_k$. We then pick the response $y$ with the highest score variance for each prompt $x$ to be used in the judge training. If multiple responses have the same variance, we break ties randomly.

**Pairwise Meta-Judge Evaluations:** For each selected response $y$, we have up to $N$ corresponding judgments, denoted as $\{j^1, \ldots, j^N\}$. We then evaluate each pair of different judgments $(j^m, j^n)$ using a meta-judge prompt shown in Figure 2. This *LLM-as-a-Meta-Judge* prompt includes the original prompt $x$, response $y$, and its two judgements $(j^m, j^n)$ as well as the rubric used by the judge. Then the model is asked to generate chain-of-thought reasoning followed by its choice of the better judgement. Again this uses the same LLM model, but acting as a meta-judge this time.

To mitigate positional bias , we prompt the model twice by changing the ordering of the two judgements. In addition, we also introduce weighted scoring for winning in the first vs second positions. We define $win_{1st}$ and $win_{2nd}$ as the total wins in the first and second positions respectively, and calculate the weights as:

$$\omega_1 = \frac{win_{2nd}}{win_{1st} + win_{2nd}}, \qquad \omega_2 = \frac{win_{1st}}{win_{1st} + win_{2nd}}.$$

The result of a single battle between judgments $(j^m, j^n)$ is defined as:

$$r^{mn} = \begin{cases} 1 & \text{If the meta-judge prefers } m \text{ wins} \\ -1 & \text{If the meta-judge prefers } n \text{ wins} \\ 0 & \text{If tie or parse error.} \end{cases}$$

We then construct a battle matrix $B$ as the weighted sum of the battle results:

$$B_{mn} = \omega_1 \mathbf{1}[r^{mn} = 1] + \omega_2 \mathbf{1}[r^{nm} = -1]$$

**Elo Score and Pairs Selection:** The next step is to convert the battle matrix into rewards (meta-rewards) corresponding to each judgement. Inspired by Zheng et al. (2024), we determine the Elo score $\varepsilon_m$ for each judgment $j^m$ by solving the following maximum likelihood estimation problem:

$$\arg\max_\varepsilon \sum_{m,n} B_{mn} \log\left(\frac{e^{\varepsilon_m - \varepsilon_n}}{1 + e^{\varepsilon_m - \varepsilon_n}}\right).$$

This approach allows us to compute scores that account for the positional bias in the meta-judge evaluations. When creating the preference pairs, we select the chosen $j^c$ and rejected $j^r$ as the judgment with the highest and lowest Elo score respectively, breaking ties randomly[3].

## 3 EXPERIMENTS

### 3.1 EXPERIMENTAL SETUP

We use instruction-finetuned Llama-3-8B-Instruct as a seed model, and otherwise closely follow the experimental setup of Yuan et al. (2024c). Before our Meta-Rewarding training, we first perform supervised finetuning (SFT) of the seed model on the Evaluation Fine-Tuning (EFT) dataset from Yuan et al. (2024c). This dataset is built from Open Assistant (Köpf et al., 2024) and provides initial LLM-as-a-Judge training data of ranked human responses, thus aiding the model to act as a judge. We refer to this model as *SFT on EFT*, or simply SFT for short.

For Meta-Rewarding iterations, we utilize 20,000 prompts from Yuan et al. (2024c) that were generated by Llama-2-70B-Chat using an 8-shot prompt. We provide a visualization of their distribution in Appendix Figure 6. For each iteration, we sample 5,000 prompts from this seed set and conduct four iterations in total. The iterative process is formally defined as follows:

Iter 1  Obtain $M_1$ by training using DPO (initialized from the SFT model) on both actor and judge preference pairs generated by the SFT model.

Iter 2  Obtain $M_2$ by training $M_1$ using DPO on actor and judge preference pairs generated by $M_1$.

Iter 3  Obtain $M_3$ by training $M_2$ using DPO exclusively on actor preference pairs generated by $M_2$.

---

[3]We perform additional length filtering after this step to mitigate length explosion

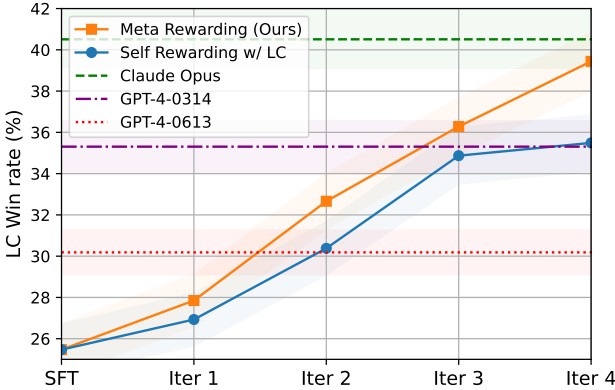

Figure 3: **AlpacaEval 2.** Length-controlled (LC) win rate increases with Meta-Rewarding iterations, even approaching Claude-Opus level. The Self-Rewarding w/LC baseline lags behind in later iterations due to its lack of judge training.

Iter 4  Obtain $M_4$ by training $M_3$ using DPO exclusively on actor preference pairs generated by $M_3$.

We provide a detailed recipe for training in Section A.5. In each iteration, we generate $K = 7$ response variations per prompt and $N = 11$ judgements per response using temperature 0.8 and top_p 0.95. We filtered out identical responses.

## 3.2 EVALUATION METHODS

As Meta-Rewarding aims to improve the model both as an actor and a judge, we evaluate its performance in both of these roles.

**Actor's Instruction Following** We make use of three well-established auto-evaluation benchmarks based on GPT4-as-a-Judge: AlpacaEval 2 (Dubois et al., 2024a), Arena-Hard (Li et al., 2024) and MT-Bench (Zheng et al., 2024). These benchmarks focus on different aspects of the model. For instance, AlpacaEval mainly focuses on chat scenarios, where the prompt sets cover a diverse range of daily questions. In comparison, Arena-Hard consist of more complex or challenging questions, where they satisfy more criteria in the predefined 7 aspects (creativity, complexity, problem-solving, etc). Notably, Arena-Hard has the highest correlation with Chatbot-Arena among popular open-ended LLM benchmarks (Li et al., 2024). MT-Bench has 8 different question categories and evaluates the multi-turn conversation ability of the model.

**Judge's Reward Modeling** To evaluate the reward modeling capability of the judge, we measure the correlation of our judge scores with human preferences, as well as a strong AI judge when human labeling is not available. We quantitatively calculate the Spearman correlation and agreement between the model-generated ranking with the human-labeled preferences provided in the Open Assistant dataset. We use a held-out split of 190 samples, with each sample consisting of a prompt and several human ranked responses, totalling 580 different responses. Additionally, we also measure the judge's performance on ranking responses generated by the seed model, which is considered to be more in-distribution compared to human or other model generated responses. This is because the judge is mainly trained and applied on samples that are self-generated. However, in this case, we do not have ground-truth human preference labels, so we adopt the strong judge gpt-4-1106-preview as a proxy.

## 3.3 INSTRUCTION FOLLOWING EVALUATION

**Meta-Rewarding iterations significantly improves the win rate.** In Figure 3, we show the length-controlled (LC) win rate of our method over its training iterations on the AlpacaEval benchmark. Overall, we see a substantial increase from 22.9% to 39.4%, outperforming GPT-4 and approaching close to the Claude Opus model. This is a remarkable result considering our model has only 8B parameters and our training did not utilize any extra human data beyond the seed model (except the

---

[4]Our evaluation shows slightly higher numbers, with the LC Winrate 24.57%, Winrate 24.89% and Length 1936. This is likely due to a different inference template.

Table 1: **AlpacaEval 2:** The evaluation on AlpacaEval shows significant improvement with Meta-Rewarding training. While the seed model Llama-3-8B-Instruct only achieves 22.92% length-controlled (LC) win rate against GPT4-Turbo, our 4-th iteration achieves 39.44%.

| Model | LC win rate | Win rate | Length |
|---|---|---|---|
| Llama-3-8B-Instruct (Seed)[4] | 22.92% | 22.57% | 1899 |
| SFT on EFT | 25.47% | 25.10% | 1943 |
| Self-Rewarding LLM (Yuan et al., 2024c) + LC | | | |
| *Iteration 1* | 26.93% | 27.12% | 1983 |
| *Iteration 2* | 30.38% | 29.77% | 1940 |
| *Iteration 3* | 34.87% | 34.59% | 1967 |
| *Iteration 4* | 35.49% | 35.37% | 2005 |
| Meta-Rewarding LLM (Ours) | | | |
| *Iteration 1* | 27.85% | 27.62% | 1949 |
| *Iteration 2* | 32.66% | 33.29% | 2001 |
| *Iteration 3* | 35.45% | 37.24% | 2064 |
| *Iteration 4* | **39.44%** | **39.45%** | 2003 |

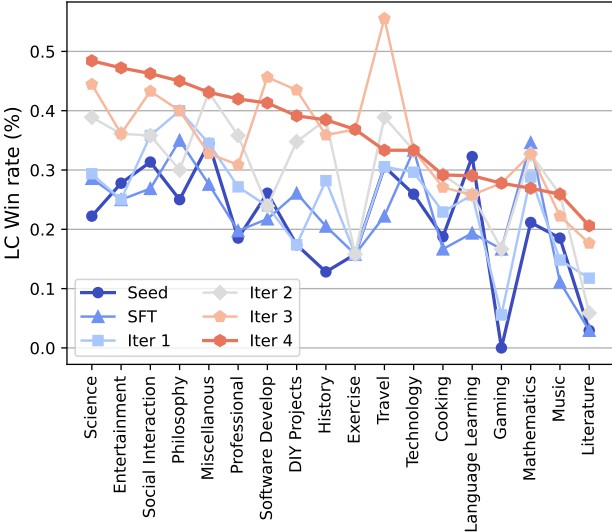

Figure 4: **Fine-grained AlpacaEval LC Winrate Analysis.** We classify all 805 AlpacaEval test prompts into 18 categories. Meta-Rewarding improves upon Llama-3-8B-Instruct for 17 out of 18 categories.

EFT dataset used in the SFT stage). In addition, our method surpasses the strong baseline of SPPO (Wu et al., 2024), which has a similar iterative training setup using Llama-3-8B-Instruct, but uses a reward model that was trained on a large set of human and GPT-4 data. Despite its reliance on a strong external reward model as a judge, SPPO achieves 38.77% LC win rate, which is slightly lower than our method.

**Importance of the meta-judge and length-control mechanism.** The Self-Rewarding baseline with our length-control (LC), which lacks the meta-judge for training the judge, also brings improvement, but to a lesser degree, especially in later iterations. This signifies the importance of training the judge and the effectiveness of the meta-judge in achieving this. As shown in Table 1, the average response length (measured in characters) does not grow substantially over training iterations, proving the effectiveness of our length-control mechanisms (see ablations in Section 3.5).

**Meta-Rewarding improves nearly all instruction categories.** We perform a fine-grained analysis by breaking down the 805 questions in AlpacaEval into 18 categories[5] given in Yuan et al. (2024c). Notably, we find significant improvements in most of the categories as shown in Figure 4, including

---

[5]We dropped 2 categories that had less than 10 samples.

Table 2: **Arena-Hard:** Although our prompt set are far from the distribution of Arena-Hard (which is selected from the highest quality clusters from the Chatbot Arena dataset), we observe a substantial improvement. Four iterations of Meta-Rewarding brings +8.5% increase over the seed model.

| Model | Score | 95% CI | Length |
|---|---|---|---|
| Llama-3-8B-Instruct (Seed) | 20.6% | (-2.0, 1.8) | 2485 |
| SFT on EFT | 24.2% | (-2.0, 1.8) | 2444 |
| Self-Rewarding LLM (Yuan et al., 2024c) + LC | | | |
| *Iteration 1* | 23.2% | (-1.7, 1.9) | 2438 |
| *Iteration 2* | 26.3% | (-2.1, 2.3) | 2427 |
| *Iteration 3* | 28.2% | (-2.0, 1.9) | 2413 |
| *Iteration 4* | 27.3% | (-2.0, 2.2) | 2448 |
| Meta-Rewarding LLM (Ours) | | | |
| *Iteration 1* | 25.1% | (-1.9, 1.8) | 2395 |
| *Iteration 2* | 27.4% | (-2.0, 2.0) | 2416 |
| *Iteration 3* | 27.6% | (-2.3, 2.6) | 2501 |
| *Iteration 4* | **29.1%** | (-2.3, 2.1) | 2422 |

categories that require a considerable amount of knowledge and reasoning, *e.g.* science, gaming, literature, etc. However, there are also categories like Travel or Mathematics, where the model only has slight improvement compared with the seed model Llama-3-8B-Instruct.

**Meta-Rewarding improves answering of complex and hard questions.** We further evaluate our method's performance on answering complex and challenging prompts using Arena-Hard. The evaluation results in Table 2 show that Meta-Rewarding is able to improve the score in all 4 iterations, showing a substantial improvement (+8.5%) compared with the seed model (20.6%). This further validate the effectiveness of our method.

### 3.4 REWARD MODELING EVALUATION

We evaluate the judging accuracy of our models on responses generated by the seed model Llama-3-8B-Instruct. In the absence of human labeling, we measure the correlation between our model and the currently strongest judge model, gpt-4-1106-preview. Our analysis employs two slightly different settings, primarily differing in how they handle ties given by the judge models. We begin with a fixed set of Open Assistant prompts that do not overlap with our training prompts.

For the *GPT-4 Chosen Pairs* setting, we generate two responses using the seed model for each prompt. We then generate preference labels with GPT-4 judge using a prompt adopted from AlpacaEval (see Section A.1). To mitigate positional bias, we make two judgements by switching the positions of the compared responses. We retain the data only where the two judgments agree, discarding the rest. This process yields a total of 170 pairs with preference judge labels. Subsequently, we use the model being evaluated to predict rankings on those pairs, employing the same procedure as before by generating 11 judgments and averaging their scores. We calculate two metrics: agreement (counting ties as 0.5) and agreement without ties (removing all ties predicted by the weaker judge and assessing agreement on the remaining pairs).

For the *Self-Chosen Pairs* setting, we generate 7 responses from the seed model and rank them using the target model. Again, we use the same procedure of averaging of 11 judgements. We select the highest and lowest scoring responses as the predicted chosen and rejected pairs, respectively. We then perform the same judgment using the strong GPT-4 model and report the agreement and agreement without ties metrics.

**The model improves in judging after performing judge training:** Our analysis shown in Table 3 reveals significant improvements in the correlation between Meta-Rewarding and the strong GPT-4 judge compared to the Self-Rewarding baseline in both evaluation settings. The enhancement is most notable in the agreement without ties metric. For *Self-Chosen Pairs*, the improvement reaches up to +12.34% (*Iteration 2*) when comparing the same iterations of both models, while in the *GPT-4 Chosen Pairs* setting, the increase exceeds +6%.

Table 3: **Judge agreement with GPT-4 on responses generated by the seed model:** Evaluation of the judge's correlation with GPT4 on the Open Assistant test set, with responses generated by Llama-3-8B-Instruct.

| Model | GPT-4 Chosen Pairs | | Self-Chosen Pairs | |
|---|---|---|---|---|
| | Agreement | Agree wo Tie | Agreement | Agree wo Tie |
| Llama-3-8B-Instruct (Seed) | 55.95% | 56.49% | 55.80% | 61.03% |
| SFT on EFT | 51.48% | 51.79% | 61.66% | 73.51% |
| Self-Rewarding LLM (Yuan et al., 2024c) + LC | | | | |
| *Iteration 1* | 56.54% | 57.97% | 55.17% | 59.59% |
| *Iteration 2* | 52.67% | 53.43% | 54.89% | 60.00% |
| *Iteration 3* | 55.65% | 55.90% | 61.13% | 72.68% |
| *Iteration 4* | 52.97% | 53.12% | 64.44% | 78.42% |
| Meta-Rewarding LLM (Ours) | | | | |
| *Iteration 1* | 56.54% | 57.23% | 60.06% | 68.75% |
| *Iteration 2* | 55.05% | 56.58% | 61.57% | 72.34% |
| *Iteration 3* | **58.63%** | **61.24%** | 63.43% | 76.80% |
| *Iteration 4* | 57.44% | 59.54% | **64.50%** | **79.33%** |

Table 4: **Effect of Length-Control Parameter $\rho$ on AlpacaEval:** We find that the length-control parameter $\rho$ significantly impacts both the win rate and length-controlled (LC) win rate. Using a larger threshold decreases the model generation length, and vise versa. While turning off the length-control mechanism ($\rho = 0$) increases the win rate, it hurts the LC win rate and makes the responses longer. Choosing a balanced length-control parameter provides a balanced final performance.

| Model | LC win rate | Win rate | Length |
|---|---|---|---|
| Self-Rewarding LLM + LC | | | |
| *Iteration 3 (Base)* | 34.87% | 34.59% | 1967 |
| *Iteration 4 ($\rho = 0$)* | 34.68% | **36.11%** | 2063 |
| *Iteration 4 ($\rho = 0.1$)* | 35.49% | 35.37% | 2005 |
| *Iteration 4 ($\rho = 0.3$)* | **35.83%** | 31.95% | 1806 |
| Meta-Rewarding LLM (Ours) | | | |
| *Iteration 3 (Base)* | 35.45% | 37.24% | 2064 |
| *Iteration 4 ($\rho = 0$)* | - | - | 2212 |
| *Iteration 4 ($\rho = 0.3$)* | - | - | 2127 |
| *Iteration 4 ($\rho = 0.35$)* | - | - | 2067 |
| *Iteration 4 ($\rho = 0.4$)* | **39.44%** | **39.45%** | 2003 |

**Meta-Rewarding training improve judging correlation with Human.** We examine the judge's correlation with the human-ranked responses from the Open Assistant dataset. As shown in Appendix Table 7, we measure the agreement as well as the average Spearman correlation (over prompts). There is a notable increase in correlation with human judgement, especially in Meta-Rewarding LLMs.

## 3.5 ABLATIONS AND ANALYSIS

**Length-Control Mechanism:** Our length-control mechanism is essential in maintaining a balance between comprehensiveness and conciseness of the model responses. We compare the last training iteration with different length-control parameter choices $\rho$ and present the results in Table 4. Using $\rho = 0$ is equivalent to not performing any length-control in the preference data selection. As expected, training this way makes the model excessively verbose for both models, and negatively affects the LC win rate as shown for Self-Rewarding LLMs.

**Training with an External Reward Model:** Meta-Rewarding employs an LLM-as-a-Judge prompt to judge its own responses. Instead, we experiment with using a strong external reward model Starling-RM-34B (Zhu et al., 2023) to select actor preference pairs. However, we find that Starling-RM-34B failed to increase the LC win rate of AlpacaEval in the first iteration (24.63% vs 27.85%), perhaps due to its length bias.

Table 5: **Meta-Judge Statistics.** We observe growing biases in the meta-judge towards preferring higher score judgements or those in the first position.

| Meta-Judge | Score Bias | | Positional Bias | | |
|---|---|---|---|---|---|
| | Higher Win | Lower Win | Same Score | Diff Score | All |
| *Iteration 1* | 63.04% | 36.96% | 47.79% | 41.12% | 43.92% |
| *Iteration 2* | 97.68% | 2.32% | 87.75% | 56.18% | 68.11% |

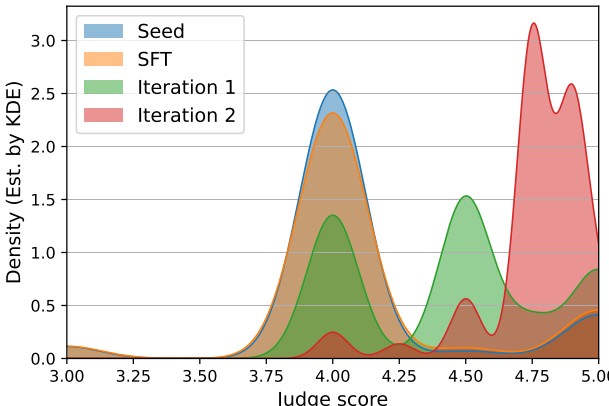

Figure 5: **Change in Scoring Distribution:** Training the judge using the meta-judge changes its score distribution significantly. Notably, the judge tends to concentrate more into giving a high score. As a result, the mean score is increased from 4.1 to 4.7+ after two iterations of training.

**Meta-Judge Biases:** After the first iteration of Meta-Rewarding training, the meta-judge becomes more likely to prefer a higher score judgment nearly all the time, as shown in Table 5. This score-bias, in turn, significantly shifts the scoring distribution of the judge towards the full score of 5. For the positional bias, we also see an increasing trend of during the training, especially for comparing two judgments with the same score.

**Judge Scoring Shift.** To investigate the judge score distribution change during Meta-Rewarding training iterations, we use the same validation prompts as used for reward modeling evaluation. We generate 7 responses on each prompt using Llama-3-8B-Instruct, then generate 11 judgments for each response. Figure 5 is a visualization of the scoring distribution, where the density is estimated using Gaussian kernel density estimation (Davis et al., 2011). Training the judge using the meta-judge further increases its likelihood of generating higher scores. However, we notice that the first 2 iterations of the judge training makes it prefer to assign scores 4.5, 4.75, 4.9 even though the scores should be integers according to the instruction. Although these are high scores, they provide more granularity and distinguishing ability for separating different quality responses.

## 4 RELATED WORK

**RLHF** Alignment strategies can be broadly classified into aligning with a reward model or aligning directly based on a preference dataset. Ziegler et al. (2019); Stiennon et al. (2020); Ouyang et al. (2022); Bai et al. (2022a) train a fixed reward model from human preference data, and then use the reward model to train via reinforcement learning (RL), e.g. via Proximal Policy Optimization (PPO) (Schulman et al., 2017). To further reduce engineering costs, P3O (Wu et al., 2023) derived the contrastive policy gradient, which has shown superior performance over PPO while removing the need for a value function. In contrast, methods such as Direct Preference Optimization (DPO) (Rafailov et al., 2024; Xu et al., 2023; Zhao et al., 2023; Zheng et al., 2023; Yuan et al., 2024a) avoid training the reward model entirely, and instead directly train the LLM using human preferences.

**LLM-as-a-Judge** Using LLM-as-a-Judge for evaluation (Li et al., 2024; Dubois et al., 2023; 2024b; Saha et al., 2023; Bai et al., 2024) and training reward models (Lee et al., 2023; Zhu et al., 2023;

Chen et al., 2023; Li et al., 2023) has become a standard practice. Some works, such as Kim et al. (2023; 2024), have investigated how to construct datasets for training a LLM-as-a-Judge. However, these approaches typically use human data or data coming from a much stronger model. In contrast, our approach emphasizes self-improvement of judgment skills.

**Super Alignment** Since current alignment methods mostly rely on either supervised fine-tuning (SFT) with human-provided demonstrations (Sanh et al., 2021; Wei et al., 2021; Chung et al., 2024) or reinforcement learning from human feedback (RLHF) (Ziegler et al., 2019; Stiennon et al., 2020; Ouyang et al., 2022), their capabilities would be inherently limited as humans cannot always provide helpful demonstrations or supervision on the hard tasks beyond their expertise (Sharma et al., 2023). Several promising directions toward super alignment exist, including using models to assist human supervision (Bowman et al., 2022; Saunders et al., 2022; Leike et al., 2018; Lightman et al., 2023), automatic search for problematic behaviors or internals (Perez et al., 2022; Bills et al., 2023; Templeton, 2024) and more. The closest direction to our work is using AI to produce feedback for training AI, also known as RLAIF (Zhu et al., 2023; Lee et al., 2023). For example, Constitutional AI (Bai et al., 2022b) uses an LLM to provide feedback and refine responses.McAleese et al. (2024) trained CriticGPT to write critiques that highlight inaccuracies in ChatGPT answers. Self-Rewarding Yuan et al. (2024c) proposed an iterative training scheme where the model acts as a judge to evaluate its own responses and then that feedback is used in the preference optimization. However, as far as we know, less work has focused on training the actor and the judge simultaneously during self-improvement.

## 5 LIMITATIONS

A deficiency in our experimental setup is the 5-point judging system that we chose, following Yuan et al. (2024b). We discovered that this scoring method often results in ties due to minimal quality differences between responses, necessitating careful averaging of multiple judgments to differentiate between them. Moreover, as training progressed, responses increasingly approached the maximum score, making further improvements difficult to detect. A more nuanced scoring system that covers diverse aspects (Wang et al., 2024) or a comparison-based approach might address these issues.

Another significant limitation lies in the judge training process. Despite our efforts to mitigate positional bias of our *meta-judge*, this issue persists and hindered further improvements in *Iteration 3*. The judge also demonstrated a tendency to assign higher scores, which accelerated score saturation and reduced its ability to discriminate between responses.

## 6 CONCLUSION

In this work, we propose a novel mechanism for improving the judging skill of models by using a *meta-judge* that assigns *meta-rewards* to select chosen and rejected judgments for preference optimization. This addresses a major limitation of the Self-Rewarding framework (Yuan et al., 2024c), specifically the lack of training the judge. We additionally introduce a new length-control technique to mitigate the issue of length explosion when training with AI feedback. The effectiveness of our method is demonstrated through auto-evaluation benchmarks AlpacaEval, Arena-Hard, and MT-Bench. Remarkably, even without additional human feedback, our approach significantly improves upon Llama-3-8B-Instruct and surpasses both Self-Rewarding and SPPO (Wu et al., 2024), a strong baseline that relies heavily on human feedback. Furthermore, when we evaluate our model's judging ability, it shows significant improvement in correlation with both human judges and strong AI judges like gpt-4-1106-preview. Overall, our findings provide strong evidence that self-improving the model without human feedback is a promising direction for achieving super alignment.

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

# A APPENDIX

## A.1 JUDGE PROMPT

---

**Pointwise Judge Prompt**

Review the user's question and the corresponding response using the additive 5-point scoring system described below. Points are accumulated based on the satisfaction of each criterion:

- Add 1 point if the response is relevant and provides some information related to the user's inquiry, even if it is incomplete or contains some irrelevant content.
- Add another point if the response addresses a substantial portion of the user's question, but does not completely resolve the query or provide a direct answer.
- Award a third point if the response answers the basic elements of the user's question in a useful way, regardless of whether it seems to have been written by an AI Assistant or if it has elements typically found in blogs or search results.
- Grant a fourth point if the response is clearly written from an AI Assistant's perspective, addressing the user's question directly and comprehensively, and is well-organized and helpful, even if there is slight room for improvement in clarity, conciseness or focus.
- Bestow a fifth point for a response that is impeccably tailored to the user's question by an AI Assistant, without extraneous information, reflecting expert knowledge, and demonstrating a high-quality, engaging, and insightful answer.

User: {query}

<response>{response}</response>

After examining the user's instruction and the response:

- Briefly justify your total score, up to 100 words.
- Conclude with the score using the format: "Score: <total points>"

Remember to assess from the AI Assistant perspective, utilizing web search knowledge as necessary.

---

We adopt the same judge prompt as in Yuan et al. (2024c).

## A.2 META-JUDGE PROMPT

---

**LLM-as-a-Meta-Judge Prompt**

Review the user's question and the corresponding response, along with two judgments. Determine which judgment is more accurate according to the rubric provided below. The rubric used for the initial judgments is as follows:

- Add 1 point if the response is relevant and provides some information related to the user's inquiry, even if it is incomplete or contains some irrelevant content.
- Add another point if the response addresses a substantial portion of the user's question, but does not completely resolve the query or provide a direct answer.
- Award a third point if the response answers the basic elements of the user's question in a useful way, regardless of whether it seems to have been written by an AI Assistant or if it has elements typically found in blogs or search results.
- Grant a fourth point if the response is clearly written from an AI Assistant's perspective, addressing the user's question directly and comprehensively, and is well-organized and helpful, even if there is slight room for improvement in clarity, conciseness or focus.
- Bestow a fifth point for a response that is impeccably tailored to the user's question by an AI Assistant, without extraneous information, reflecting expert knowledge, and demonstrating a high-quality, engaging, and insightful answer.

User: {prompt}

Response:
{response}

Judgment A:
{judgment_a}

Judgment B:
{judgment_b}

After examining the original question, response, and both judgments:

- Explain which judgment is more accurate according to the original rubric and why. Consider factors such as adherence to the rubric, accuracy in evaluating the response, and consistency in applying the criteria.
- Conclude with a clear statement of which judgment is better using the format: "Winner: [Judgement A | Judgement B]"

---

## A.3 GPT4 JUDGE PROMPT

---

**alpaca_eval_clf_gpt4_turbo**

<|im_start|>system
You are a highly efficient assistant, who evaluates and selects the best large language model (LLMs) based on the quality of their responses to a given instruction. This process will be used to create a leaderboard reflecting the most accurate and human-preferred answers.
<|im_end|>
<|im_start|>user
I require a leaderboard for various large language models. I'll provide you with prompts given to these models and their corresponding outputs. Your task is to assess these responses, and select the model that produces the best output from a human perspective.

## Instruction

{
    "instruction": """{instruction}""",
}

## Model Outputs

Here are the unordered outputs from the models. Each output is associated with a specific model, identified by a unique model identifier.

{
    {
        "model_identifier": "m",
        "output": """{output_1}"""
    },
    {
        "model_identifier": "M",
        "output": """{output_2}"""
    }
}

## Task

Evaluate the models based on the quality and relevance of their outputs, and select the model that generated the best output. Answer by providing the model identifier of the best model. We will use your output as the name of the best model, so make sure your output only contains one of the following model identifiers and nothing else (no quotes, no spaces, no new lines, ...): m or M.

## Best Model Identifier
<|im_end|>

---

We adopt this prompt from AlpacaEval, which is proved to have high correlation with human judges.

## A.4 MT-BENCH RESULTS

**Meta-Rewarding does not sacrifice multi-turn ability despite training only on single-turn.** We perform MT-Bench evaluation to examine the loss in multi-turn conversation ability since we trained only on single-turn data. The result (detailed in Appendix Table 6) shows that Meta-Rewarding significantly improves the Turn 1 Score from 8.319 to 8.738 in the last iteration, while sacrificing no more than 0.1 in Turn 2 Score. This is a large improvement on Self-Rewarding + LC, as it typically sacrifices more than 0.2 in Turn 2 score while not improving the Turn 1 score.

## A.5 TRAINING DETAILS

For the SFT model, we train for a total of 10 epochs using a learning rate $5 \times 10^{-8}$ and global batch size of 32. We employed cosine learning rate scheduling and saved a checkpoint after every epoch. We selected checkpoint from epoch 5 as the final model.

For all DPO training, we also trained for 10 epochs, with a learning rate of $5 \times 10^{-6}$, $\beta = 0.1$ and global batch size of 32. We adopted cosine learning rate scheduling.

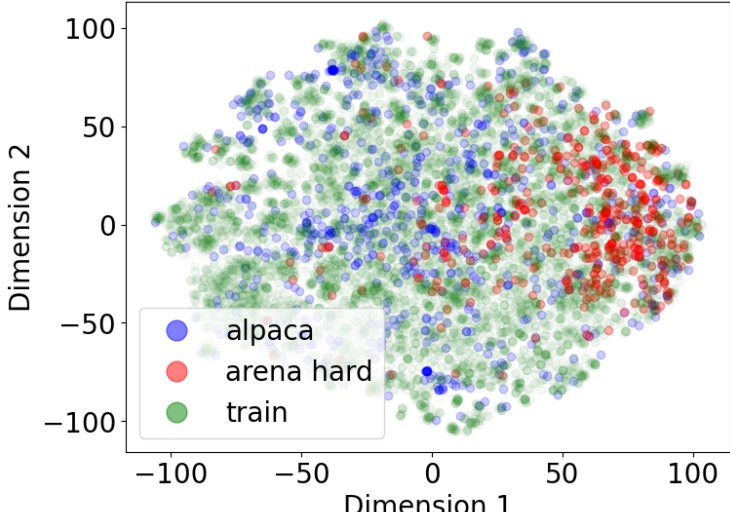

Figure 6: **Distribution of Prompts:** A t-SNE to visualization of three sources of prompts: training prompts, AlpacaEval prompts and Arena-Hard prompts. The embedding of the prompts are calculated by text-embedding-3-small. Our training prompts are closer in distribution to AlpacaEval prompts, while Arena-Hard is more concentrated into a subset of the distribution.

Table 6: **MT-Bench:** Since our training mainly focus on the first-turn capability, we observe a significant improvement in the Turn 1 Score. While the Self-Rewarding baseline suffer from a large drop in Turn 2 score, our Meta-Rewarding only sacrifice slightly and even improving the Turn 2 score in *Iteration 3 & 4*.

| Model | Score | Turn 1 | Turn 2 | Length |
|---|---|---|---|---|
| Llama-3-8B-Instruct | 8.116 | 8.319 | 7.911 | 1568 |
| SFT on EFT | 7.943 | 8.138 | 7.747 | 1511 |
| Self-Rewarding LLM + LC | | | | |
| *Iteration 1* | 7.909 | 8.144 | 7.671 | 1576 |
| *Iteration 2* | 7.894 | 8.200 | 7.588 | 1570 |
| *Iteration 3* | 7.984 | 8.231 | 7.734 | 1528 |
| *Iteration 4* | 8.028 | 8.381 | 7.675 | 1539 |
| Meta-Rewarding LLM | | | | |
| *Iteration 1* | 7.994 | 8.263 | 7.725 | 1555 |
| *Iteration 2* | 8.198 | **8.794** | 7.595 | 1577 |
| *Iteration 3* | **8.341** | 8.731 | **7.950** | 1596 |
| *Iteration 4* | 8.288 | 8.738 | 7.838 | 1592 |

For Self-Rewarding training, during Iteration 1 we set $\rho = 0$ for actor data creation and applied a filter to exclude pairs where the chosen response length exceeded 2500 characters. We selected the checkpoint from epoch 5 for this iteration. In both Iteration 2 & 3 we continue with $\rho = 0$ and chose checkpoints from epoch 1 and epoch 2 respectively. For Iteration 4, we adjust $\rho$ to 0.1 and selected the checkpoint from epoch 2.

For Meta-Rewarding training in Iteration 1 we set $\rho = 0$ for actor data creation, and we filtered out pairs with chosen response length exceeding 2500 characters. Additionally, for the judge data creation, we filtered out pairs if the chosen judgment length exceeded 1100. We selected checkpoint from epoch 6 for this iteration. In Iteration 2, we increased $\rho$ to 0.32 and set the threshold to 1000 for judge data filtering, we selected the checkpoint from epoch 4. In Iteration 3 we maintain $\rho$ at 0.32 and chose the checkpoint from epoch 2. Finally, in Iteration 4, we further increased $\rho$ to 0.4 and again selected the checkpoint from epoch 2.

Table 7: **Judge's Correlation with Human:** We measure the judge's agreement (with and without ties) with humans on the Open Assistant test set. Spearman correlation represent the ranking spearman correlation with the ground truth averaged over prompts.

| Model | Agreement | Agree wo Tie | Spearman corr. |
|---|---|---|---|
| Llama-3-8B-Instruct | 62.81% | 64.18% | 0.315 |
| SFT on EFT | 63.20% | 64.59% | 0.321 |
| Self-Rewarding LLM + LC | | | |
| *Iteration 1* | 63.04% | 65.04% | 0.298 |
| *Iteration 2* | 64.14% | 67.17% | 0.347 |
| *Iteration 3* | 60.23% | 61.63% | 0.251 |
| *Iteration 4* | 61.48% | 62.22% | 0.283 |
| Meta-Rewarding LLM | | | |
| *Iteration 1* | 57.73% | 61.98% | 0.210 |
| *Iteration 2* | **66.64%** | **68.33%** | **0.382** |
| *Iteration 3* | 63.35% | 65.24% | 0.329 |
| *Iteration 4* | 62.96% | 64.82% | 0.326 |