# OpenReview forum: "Meta-Rewarding Language Models: Self-Improving Alignment with LLM-as-a-Meta-Judge"
_ICLR.cc/2025/Conference — Submitted to ICLR 2025_

### Official Review · Reviewer_Kg4s · 2024-10-26

**Soundness:** 2
**Presentation:** 3
**Contribution:** 2
**Rating:** 3
**Confidence:** 4

**Summary:**

The paper proposed Meta-rewarding, an additional layer on synthetic data generation that uses judgments on (model’s) prior judgments in order to improve the quality of synthetic data generation and hence improve the quality of the aligned model. This added verification layer sits on top of the model's generation ability to mitigate saturation in “self-rewarding mechanisms”.
The paper uses  Llama-3-8B-Instruct for evaluation and conducts evaluation with length-normalized AlpachaEval 2. The evaluation shows that the proposed approach improves upon the prior work by Yuan et al.

**Strengths:**

- Overall, the idea is imaginative and interesting.  The method is simple and intuitive.
- The performance increase is large (22.9% → 39.4%) on AlpacaEval 2.0 LC.
- The paper is clearly written, with details on the data construction and synthesis, making the work reproducible. Overall, the writing is smooth, and I was able to follow most of the ideas (there are places which can be improved, as I discuss later on).

**Weaknesses:**

- The proposed approach requires more computation (compared to their baseline Yuan et al). Hence, if the comparison is for a fixed compute budget, which is the best data augmentation method, one needs to redo the experiments (e.g., Fig3) to obtain compute-adjusted comparison results. I acknowledge that it's fair to control for "iteration" only if the goal is to solve the saturation problem in data augmentation, but that’s a different goal.
- The entire setup of self-rewarding only assumes no human annotator available, but ignores the fact that you can take off-the-shelf reward models to rank your responses. It would be useful to compare self-rewarding with LLM-judge and iterative DPO with rejection sampling / Best-of-N sampling, which I believe is a more practical setting that is adopted by many works [1, 2]. If you can take off-the-shelf SFT models, and assume “no human annotator available”, why can’t you take off-the-shelf reward models under the same assumption.
- The setup assumes that the base LM is bad at judging (or otherwise, you wouldn’t need to train the judging ability) but good at meta-judging (since you take the meta-judge results as gold results to train the judge) - which might be true for LLama-3-8B-Instruct. This work only investigated ONE base model (llama3-8b-inst) on TWO different benchmarks (Arena Hard, Alpaca Eval 2.0), therefore it is unclear if the assumption still holds for other models / benchmarks. As the LLama3 Instruct model’s training data is unknown - it is unclear whether the improvements comes from the proposed pipeline or simply because LLama 3 instruct is trained on human annotated meta judging data - which contradicts the setup of “without human labelers”.
- It is unclear what is the upper limit of self-rewarding [3] and meta-rewarding (this work) as the former only reports performance up to 3 iterations and this work only reports performance up to 4 iterations. If, after these iterations, the performance stops increasing, then this should also be reported so that the community can learn about the upper bound of self-rewarding mechanisms. The authors claim that the proposed method solves the “rapid saturation” problem in [3], but extending from 3 iters to 4 can hardly be viewed as solving “rapid saturation”.

References

- [1] The Llama3 Herd of Models (Dubey et al., 2024)
- [2] Iterative Preference Learning from Human Feedback: Bridging Theory and Practice for RLHF under KL-constraint (Xiong et al., 2023)
- [3] Self-Rewarding Language Models (Yuan et al., 2024)

**Questions:**

- Last equation  before S3: $\epsilon_m$ and $\epsilon_n$ are two distinct dimensions of $\epsilon$? I don't see where you defined $\epsilon$. Make it clear.


Missing citations: Here are several works that are not cited in your work:
- Crystal: Introspective reasoners reinforced with self-feedback, 2023
- Generating sequences by learning to self-correct, 2023
- The self-Instruct paper, 2023
- The Reflexion paper, 2023


Suggestion: There is also literature that casts doubt on the abilities of LLMs to self-improve out of the box. It would be great to also expand on this literature:
- The DeepMind paper: Large language models cannot self-correct reasoning yet, 2024
- Self-[In]Correct: LLMs Struggle with Discriminating Self-Generated Responses, 2024
- The Google paper: LLMs cannot find reasoning errors, but can correct them given the error location, 2024
- ... (there are a few others that I might be missing)


Your work is also related to the growing literature on inference-scaling. As the prior work has shown, inference-scaling works. Naturally, one can extend these to data generation, as you do in your work.


Finally, there is prior work on alignment of various sorts that also does length-controlled preference data selection (https://arxiv.org/pdf/2402.07319, https://arxiv.org/abs/2404.03862); highlighting for your attention. These may be hidden in experimental details since, typically, no one makes a big deal of their length-controlled data filtering/selection.

---

> ### Author Response · Authors · 2024-11-19
>
> We thank the reviewer, but we respectively disagree with the main rejection reasons. See our responses below.
>
> > The proposed approach requires more computation (compared to their baseline Yuan et al). Hence, if the comparison is for a fixed compute budget, which is the best data augmentation method, one needs to redo the experiments (e.g., Fig3) to obtain compute-adjusted comparison results. I acknowledge that it's fair to control for "iteration" only if the goal is to solve the saturation problem in data augmentation, but that’s a different goal.
>
> This is not accurate. Spending more compute in Self-Rewarding LLM does not necessarily lead to performance improvement. This is evidenced by the saturation observed in Self-Rewarding LLM in Figure 3, which shows that while we may aim to use more compute, it does not result in better outcomes. Even within each iteration, we rarely use the final checkpoint during checkpoint selection, indicating that allocating more compute for training Self-Rewarding LLM often leads to overfitting rather than improved performance. In contrast, Meta-Rewarding offers a more effective way to utilize additional compute per sample, significantly enhancing performance and mitigating the saturation problem.
>
> > If you can take off-the-shelf SFT models, and assume “no human annotator available”, why can’t you take off-the-shelf reward models under the same assumption.
>
> Using an external reward model is a well-studied research problem but is outside the scope of our paper. Our method assumes a self-improving training setup where no external reward model or human data is available. Training an external reward model requires human annotation, which is costly to obtain. Furthermore, our setup is closely related to the challenge of superalignment, where the model reaches a regime in which humans can no longer provide reliable feedback. In such a regime, distilling a reward model from human feedback becomes infeasible. As a result, relying on off-the-shelf reward models inherently risks degrading the model's performance.
>
> > The setup assumes that the base LM is bad at judging (or otherwise, you wouldn’t need to train the judging ability) but good at meta-judging
>
> It is incorrect to state that our assumption is that the base LM is "bad" at judging—especially since "bad" is difficult to define objectively. Instead, we assume that the model makes mistakes, which is a reasonable and realistic premise. We further assume that, when the model compares two judgments, it can identify some of these mistakes. The Meta-Judge operates based on a detailed rubric, evaluating each aspect of the judgment according to specific criteria, which makes it easier to detect flaws. Additionally, multiple judgments are compared in this way and ranked using the ELO metric, further improving the accuracy of the evaluations.
>
> > It is unclear what is the upper limit of self-rewarding [3] and meta-rewarding (this work) as the former only reports performance up to 3 iterations and this work only reports performance up to 4 iterations. If, after these iterations, the performance stops increasing, then this should also be reported so that the community can learn about the upper bound of self-rewarding mechanisms.
>
> We have not performed more than four iterations due to various resource limitations. Of course, we do not expect performance to improve indefinitely. However, we have demonstrated that Meta-Rewarding scales better than Self-Rewarding LLMs. This is evidenced in Figure 3, where Meta-Rewarding enables more effective use of compute resources before performance plateaus. While it is possible to determine the exact iteration where performance saturates, that number would likely depend on factors such as the specific model, training prompts, and overall setup, which may not generalize to other scenarios. The general rule of thumb is simple: continue iterating as long as there is improvement and resources allow.
>
> > Epsilon equations
>
> Right before the equation, we defined $\epsilon_n$ as the ELO score corresponding to judgement $n$. It is a scalar instead of a vector. While $\epsilon$ is the vector with $\epsilon_n$ as its elements.
>
> > Your work is also related to the growing literature on inference-scaling. As the prior work has shown, inference-scaling works.
>
> Our method is fundamentally different from inference-scaling papers (most of which were published after or concurrently with ours) because it does not increase inference-time compute.
>
> > Missing citation
>
> Thanks for the suggestions. We will take them into consideration in the related work.

---

> ### Comment · Reviewer_Kg4s · 2024-11-21
>
> > This is evidenced by the saturation observed in Self-Rewarding LLM in Figure 3
>
> The so called "saturation" in fig3 is within error margins. I am not ready to accept this as saturation with the current evidence.
> I would be convinced if you show the trends beyond iter4, say up to 10 iterations.
>
> > We have not performed more than four iterations due to various resource limitations
>
> Can you elaborate on this? What form of costs are we discussing here? (you don't discuss these in the paper) To my understanding, here "cost" does not change much from one iteration to another. Given that you had resources to run this for four iterations, why is it impossible to run this for a few more iterations?
>
> Also you completely ignored several of my comments. Am I correct that?
> > The proposed approach requires more computation (compared to their baseline Yuan et al).
>
> If so, have you tried the following suggested plot? (if you have the cost statistics, it should be pretty easy to plot this)
> > Hence, if the comparison is for a fixed compute budget, which is the best data augmentation method, one needs to redo the experiments (e.g., Fig3) to obtain compute-adjusted comparison results.
>
>
> You also did not address this comment:
> > This work only investigated ONE base model (llama3-8b-inst) on TWO different benchmarks (Arena Hard, Alpaca Eval 2.0),

---

> > ### Author Response · Authors · 2024-12-02
> >
> > > This is evidenced by the saturation observed in Self-Rewarding LLM in Figure 3
> > The so called "saturation" in fig3 is within error margins. I am not ready to accept this as saturation with the current evidence. I would be convinced if you show the trends beyond iter4, say up to 10 iterations.
> >
> > First of all, we think 4 iterations provide enough evidence of the efficacy of our method. The performance gap between our method and the baseline at iteration 4 is well beyond the error margins. The gap is 3.95%, while the usual standard error is around 1.4%. This shows that even with 4 iterations, our method is outperforming the baseline by a significant margin.
> >
> > > We have not performed more than four iterations due to various resource limitations
> > Can you elaborate on this? What form of costs are we discussing here? (you don't discuss these in the paper) To my understanding, here "cost" does not change much from one iteration to another. Given that you had resources to run this for four iterations, why is it impossible to run this for a few more iterations?
> >
> > Like any other training method, more iterations requires more time and compute resources, including cost of API access. Like any research project, we have limited resources (compute, API cost, etc) that dictates how many iterations we can perform. The main baseline of Self-Rewarding LLM does 3 iterations, which we increased by one.
> >
> > In addition to the resources constraint, we discussed in the paper limitations of our method that prevents us from performing more iterations. For example, as discussed in Sec.5, the judge score becomes saturated, making it difficult to separate chosen and rejected responses.
> >
> > > Also you completely ignored several of my comments. Am I correct that?
> >
> > We tried to answer all your questions. Please let us know if there is any specific question you want us to address.
> >
> > > The proposed approach requires more computation (compared to their baseline Yuan et al).
> > If so, have you tried the following suggested plot? (if you have the cost statistics, it should be pretty easy to plot this)
> >
> > We haven’t tried such a plot. As we mentioned, the computational cost is not the main factor in this experimental setup. Naively increasing the computational cost doesn’t improve performance. Training more epochs in each iteration using more computation actually hurts performance as the model starts to overfit to the training data.
> >
> > > You also did not address this comment:
> > This work only investigated ONE base model (llama3-8b-inst) on TWO different benchmarks (Arena Hard, Alpaca Eval 2.0),
> >
> > We apologize. We didn’t address this because it sounded more like a statement rather than a question. Yes, we used one base model and evaluated using two benchmarks. But we also did fine-grained analysis (Fig 4), reward-model evaluation (Sec 3.4), and different ablations. The two benchmarks we used are widely used and well accepted in the research community.

---

### Official Review · Reviewer_NeKw · 2024-11-04

**Soundness:** 3
**Presentation:** 3
**Contribution:** 3
**Rating:** 6
**Confidence:** 4

**Summary:**

The authors aim to improve recent self-rewarding mechanisms (Yuan et al., 2024c). Self-rewarding LLM improves the LLM response generation by judging its own responses. Typically there are two roles of LLM in the Self-rewarding setup, actor and judge. The actor generates responses towards certain prompt distribution and the same model then acts as a judge and discerns the good and bad outputs.
However, the authors recognize a key limitation of the Self-rewarding LLM approach, that it only focuses on improving the actor and not the judge. To mitigate this, they propose a 3rd role of meta-judge, which evaluates the judgments and creates preference pairs for judge training. They call their new step Meta-Rewarding i.e. labeling its own judgments. Uniquely, in their setup, the same LLM plays all three roles - actor, judge, and meta-judge.

Another issue with the Self-rewarding setup is its tendency to increase length after each improvement iteration. To fix this, authors propose a new scheme for sampling responses, that prefer shorter responses over longer ones. In the judge role, LLM assigns a score out of 5 signifying the quality of the response for the prompt. For a given set of responses for a prompt, the authors define a region around the min and max score using an interpolation hyperparameter $\rho \ge 0$. All the responses within this boundary region are considered equally good and the shortest one is selected as the chosen preference whereas the longest one is selected as the rejected preference.

Overall their training setup iteratively trains an LLM as an actor and judge such that both operations improve resulting in a process that doesn't require human labels. The actor training uses the labels from the judge role whereas the judge training uses the labels from the meta-judge role. Their evaluations on Arena-Hard and length-controlled Alpaca Evals show that the Meta-Rewarding process can show increased performance compared to Self-Rewarding while avoiding its length increase issue (i.e. responses from the final iteration of Meta-Rewarding are usually smaller than the responses from the final iteration of Self-Rewarding).

**Strengths:**

- The paper shows an interesting paradigm for language model tuning where a single model can simultaneously optimize for better response generation and also become a better judge.
- The proposed strategy to reduce length bias seems effective given their revaluation results (Table 1, 2, and 4)
- The authors provide numerous analyses and ablations related to judge alignment and fine-grained category analysis in Alpaca Eval. It is especially exciting to see that iteration 4 of their proposed Meta-Rewarding strategy improves stably in all 18 sub-categories of AlpacaEval.

**Weaknesses:**

- The meta-judge becomes heavily biased with just one iteration of training, whereby it only picks the first position or the judgment with a higher score without actually evaluation the quality of the judgment.
   - Accordingly, out of the 4 iterations of Meta-Rewarding, the judgment training is stopped in iterations 3 and 4 while only the actor is being trained.
   - There is also a change in score distribution from the judgment phase (figure 5), indicating that subsequent iterations in Meta-Rewarding will suffer from too many ties and unclear labels for actor training.
   - These limitations of the judgment process, bring serious concerns regarding the scalability of the Meta-Rewarding approach.
- (follow-up from the previous point) In Table 3, the judgment agreement increases in iterations 3 and 4, even though only the actor model is trained in these iterations. What explains this improvement? It is unclear why actor training alone would improve judge agreement.
    - Also, given the small size of the eval set and small differences between agreement rates of Self-Rewarding and Meta-Rewarding models in Table 3, I'm concerned that those results are statistically significant.

**Questions:**

- Given the failure of improvement of judgments due to biases in meta-judge, does it make more sense to have two different LLMs one that mainly focuses on improving the judge and another that focuses on only improving the actor? Perhaps training both tasks simultaneously into a single model leads to biases in Meta-Judgements.
- Even with the length conditioned training, the average length of responses keeps increasing up until iteration 3 in Tables 1 and 2. By virtue of the preference selection process, if the chosen responses are guaranteed to be smaller than rejected responses in DPO training, is there any intuition why the length still increases in Meta-Rewarding process?
- From Figure 3, it seems like the performance from Meta-Rewarding hasn't saturated on Alpaca Eval. I'm curious if authors tried more iterations of this process and if it helped.

---

> ### Author Response · Authors · 2024-11-19
>
> We appreciate your review of our paper.
>
> > These limitations of the judgment process, bring serious concerns regarding the scalability of the Meta-Rewarding approach.
>
> Yes, as demonstrated in the paper, there are areas where improvements can be made. One of them is the saturation of the judge score, which resulted from following the setup of Self-Rewarding LLMs. Using a different scoring system or adopting pairwise comparisons could help alleviate this issue. Additionally, as you pointed out, the Meta-Judge becoming biased is another challenge, which we leave as future work. We hope that addressing these issues will further enhance our method. Nevertheless, the current version of our method already demonstrates improvements at the 8B model scale and is likely to scale effectively to 70B models as well, given that Self-Rewarding LLMs have proven successful at that scale.
>
>
> > In Table 3, the judgment agreement increases in iterations 3 and 4, even though only the actor model is trained in these iterations. What explains this improvement?
>
> While the judge is not explicitly trained, it does evolve because it shares weights with the actor, which is being trained. The actor is trained on general instructions, and judging tasks can be considered a specific instance of general instruction following. This allows for positive transfer from actor learning to the judge. This phenomenon was observed in the Self-Rewarding LLM paper (Table 4), where the judge improved despite not being explicitly trained. However, as you mentioned, Table 3 is not the most reliable metric. Instead, we should focus on the model's final performance, where significant improvements are observed.
>
> > Given the failure of improvement of judgments due to biases in meta-judge, does it make more sense to have two different LLMs one that mainly focuses on improving the judge and another that focuses on only improving the actor? Perhaps training both tasks simultaneously into a single model leads to biases in Meta-Judgements.
>
> Yes, this is an interesting direction to explore. Using a separate model would mean that the Meta-Judge remains fixed due to the lack of training data for it, which could help mitigate increasing bias. Another approach could involve making the judge pairwise, similar to the Meta-Judge, so that training the judge could also contribute to reducing positional bias in the Meta-Judge. We plan to explore these directions in future work.
>
> > Even with the length conditioned training, the average length of responses keeps increasing up until iteration 3 in Tables 1 and 2. By virtue of the preference selection process, if the chosen responses are guaranteed to be smaller than rejected responses in DPO training, is there any intuition why the length still increases in Meta-Rewarding process?
>
> The increase in response length is a general phenomenon observed in the preference alignment process and is not specific to Meta-Rewarding. Researchers are actively investigating this issue, and methods like ours have been proposed to help mitigate it. It is worth noting that, in our case, the chosen responses are shorter on average; however, this does not guarantee that a chosen response will be shorter than the rejected one for every prompt.
>
> > From Figure 3, it seems like the performance from Meta-Rewarding hasn't saturated on Alpaca Eval. I'm curious if authors tried more iterations of this process and if it helped.
>
> No, we have not conducted additional iterations due to a lack of data—specifically, running out of prompts—as well as time constraints and resource limitations.

---

### Official Review · Reviewer_zYxK · 2024-11-04

**Soundness:** 3
**Presentation:** 3
**Contribution:** 3
**Rating:** 5
**Confidence:** 4

**Summary:**

The authors propose the Meta Rewarding approach, where an LLM acts as an actor, judge, and meta-judge during iterative self-evolve to enhance its abilities as the actor and judge. Through experiments, they verify that the LLM's capabilities as both actor and judge improve over multiple iterations.

**Strengths:**

1. The authors highlight the importance of improving the model’s judging ability through iterative self-evolve, which is conceptually sound.
2. They designed separate methods for Actor Data Creation and Judge Data Creation and trained the model using DPO.
3. A new length-control technique was proposed to mitigate the issue of length explosion when training with AI feedback.

**Weaknesses:**

1. A logical concern arises regarding the authors’ approach of using the LLM as a meta-judge to improve its judging ability. If so, wouldn’t the meta-judge also need enhancement? Would this necessitate a meta-meta-judge?
2. The authors compare the Meta Rewarding and Self Rewarding methods, showing performance improvements. However, I am curious if computational costs were aligned in the comparison. This includes the overhead for generating Actor Data and Judge Data and the amount of data used for training after filtering. Especially given that the Meta Rewarding method requires additional Judge Data for training, clarifying this aspect is crucial to alleviate concerns about performance gains resulting merely from increased data construction and training costs.

**Questions:**

See weaknesses.

---

> ### Author Response · Authors · 2024-11-19
>
> Thank you for reviewing our paper.
>
> > A logical concern arises regarding the authors’ approach of using the LLM as a meta-judge to improve its judging ability. If so, wouldn’t the meta-judge also need enhancement? Would this necessitate a meta-meta-judge?
>
> Not necessarily. Even if the Meta-Judge’s performance remains constant, it can still improve the judge as long as it has good accuracy. This phenomenon was also demonstrated in the Self-Rewarding paper, where the judge was not explicitly trained yet still contributed to improving the actor''s performance. Our results further support this observation, as the Meta-Judge clearly enhances overall performance. While improving the judging capability of the Meta-Judge would likely lead to even greater performance gains, we leave that for future work.
>
> > The authors compare the Meta Rewarding and Self Rewarding methods, showing performance improvements. However, I am curious if computational costs were aligned in the comparison. This includes the overhead for generating Actor Data and Judge Data and the amount of data used for training after filtering. Especially given that the Meta Rewarding method requires additional Judge Data for training, clarifying this aspect is crucial to alleviate concerns about performance gains resulting merely from increased data construction and training costs.
>
> As background information, naively increasing compute for alignment might reduce performance due to overfitting, making it more important to focus on how much compute can be used effectively before performance saturates. This is evidenced by the saturation of Self-Rewarding LLM shown in Figure 3, which indicates that while increasing compute may seem beneficial, it does not necessarily yield improvement. In contrast, Meta-Rewarding offers a better way to utilize more compute per sample, significantly improving performance and mitigating the saturation problem. Additionally, even within each iteration, we rarely use the final checkpoint during checkpoint selection, highlighting that allocating more compute for training the Self-Rewarding LLM does not improve performance, as the model begins to overfit.

---

### Official Review · Reviewer_p6CQ · 2024-11-05

**Soundness:** 3
**Presentation:** 3
**Contribution:** 3
**Rating:** 6
**Confidence:** 4

**Summary:**

This study presents an innovative Meta-Rewarding mechanism to enhance the existing self-rewarding framework. This new approach enables the model to evaluate its judgments and incorporate this meta-level feedback to improve both its acting and judging skills simultaneously. The implementation of this mechanism in the Llama-3-8B-Instruct has yielded significant improvements in both instruction-following and reward-modeling capabilities, highlighting the potential for autonomous self-improvement without direct human supervision.

**Strengths:**

1. The motivation of explicitly enhancing judging ability, which establishes the upper bound for self-improvement of LLMs, is compelling and well-founded.
2. The proposed Meta-Rewarding approach offers the advantage of improving the response generation and judging capabilities simultaneously.
3. The writing and presentation are clear and easy to read.

**Weaknesses:**

1. The primary contribution of this paper is utilizing Meta-Rewarding to address performance saturation during iterative training, however, the judge score shifting results in less training iterations of the judge model than the actor model. This limitation not only results in the diminished performance of the judging model in the final iteration (as evidenced by GPT-4 Chosen Pairs in Table 3) but also contradicts the method's intended purpose, thereby constraining the scalability of the proposed approach.
2. The paper lacks sufficient discussion of the rationale for employing the model as its meta-judge in supervising the judging model. A critical question that emerges is the underlying mechanism behind the effectiveness of this self-supervision approach.

**Questions:**

1. In the implementation of the judge model and meta-judge model, the judge model is a point-wise scorer while the meta-judge model is a pairwise evaluator. What is the underlying rationale for this differentiation?
2. What would be the implications of adopting a consistent approach across both models - either exclusively point-wise scoring or pairwise evaluation?"
3. Regarding the evaluation of reward modeling, it would be valuable to conduct a comparative analysis between the Meta-Rewarding LLM and several established reward models, including nvidia/Llama-3.1-Nemotron-70B-Reward[1], Skywork/Skywork-Reward-Llama-3.1-8B-v0.2[2], and RLHFlow/ArmoRM-Llama3-8B-v0.1[3]. This comparison could be performed across two types of benchmarks: those specifically developed in the paper and official reward model benchmarks such as RewardBench[4] and Preference Proxy Evaluations[5].
4. Recent self-improvement approaches, such as iterative DPO[6], SPPO[7], and SimPO[8], have demonstrated considerable performance gains in AlpacaEval 2.0 when utilizing external reward models (RM) for response scoring. However, this appears to contradict the experimental findings presented in Section 3.5, which suggest that the incorporation of external RMs diminishes the model's overall performance. Could the authors elaborate on this apparent discrepancy and provide more insights about these results?

[1] HelpSteer2-Preference: Complementing Ratings with Preferences

[2] Skywork-Reward: Bag of Tricks for Reward Modeling in LLMs

[3] Interpretable Preferences via Multi-Objective Reward Modeling and Mixture-of-Experts

[4] RewardBench: Evaluating Reward Models for Language Modeling

[5] How to Evaluate Reward Models for RLHF

[6] https://snorkel.ai/blog/how-snorkel-topped-the-alpacaeval-leaderboard-and-why-we-re-not-there-anymore/

[7] Self-Play Preference Optimization for Language Model Alignment

[8] SimPO: Simple Preference Optimization with a Reference-Free Reward

---

> ### Author Response · Authors · 2024-11-19
>
> We thank the reviewers for taking the time to review our paper.
>
> > The primary contribution of this paper is utilizing Meta-Rewarding to address performance saturation during iterative training, however, the judge score shifting results in less training iterations of the judge model than the actor model. This limitation not only results in the diminished performance of the judging model in the final iteration ...
>
> The intended purpose of the paper is to improve the judge’s performance, as demonstrated by the fact that adding the Meta-Judge did enhance the performance of Self-Rewarding LLMs. Compared to Self-Rewarding, the Meta-Rewarding Iter4 judge achieves approximately 5-6% better agreement on GPT-4 Chosen Pairs. The only difference between them is the addition of the Meta-Judge. However, GPT-4 agreement is not a perfect metric, as GPT-4 can make mistakes. Instead, the final instruction-following performance should be used as the primary metric, where we observe significant gains.
>
> > The paper lacks sufficient discussion of the rationale for employing the model as its meta-judge in supervising the judging model. A critical question that emerges is the underlying mechanism behind the effectiveness of this self-supervision approach.
>
> The rationale behind the Meta-Judge is similar to the LLM-as-a-Judge paradigm. Since LLMs are trained to follow general instructions, they can be prompted to perform evaluations, enabling the LLM to judge its own outputs. Similarly, LLMs can be prompted to assess the quality of their judgments, allowing the model to evaluate its own evaluations. This approach works because it breaks down the problem of determining “which judgment is better” into verifying whether each judgment adheres to the rules outlined in the detailed rubric. Analyzing different aspects of a judgment in this way makes it easier to identify flaws that may have been overlooked during the initial judgment generation. The effectiveness of this method is demonstrated in our experiments, which show significant improvements.
>
> > In the implementation of the judge model and meta-judge model, the judge model is a point-wise scorer while the meta-judge model is a pairwise evaluator. What is the underlying rationale for this differentiation?
>
> For the Meta-Judge, we found that using a pointwise judge can result in high scores that barely differentiate between judgments, providing less signals for training compared to a pairwise judge. In contrast, the lack of separability is less problematic when training the actor, so we retain the same setting as the Self-Rewarding LLM for better consistency.
>
> > What would be the implications of adopting a consistent approach across both models - either exclusively point-wise scoring or pairwise evaluation?
>
> If both models use pointwise scoring, there is not much to change in the algorithm, except that the performance may vary depending on the pointwise judging capability of the Meta-Judge. However, if both models use pairwise scoring, the Meta-Judge must evaluate a pair of "pairwise" judgments. In this case, it takes two responses, A and B, as input, along with two judgments with differing results (e.g., scores indicating A > B and A < B). The Meta-Judge then determines which judgment is better.
>
>
> > Regarding the evaluation of reward modeling, it would be valuable to conduct a comparative analysis between the Meta-Rewarding LLM and several established reward models, …
>
> The goal of this paper is to enhance self-improving training, where the model cannot rely on external models or additional data. Specifically, we focused on improving the judging capability of self-rewarding LLMs. Comparing this approach to established reward models is not entirely fair, as those models are not trained in a self-improving manner and typically depend on extra human data or outputs from stronger models, such as GPT-4.
>
> > Recent self-improvement approaches, such as iterative DPO[6], SPPO[7], and SimPO[8], have demonstrated considerable performance gains in AlpacaEval 2.0 when utilizing external reward models (RM) ...
>
> Our setup is closely related to the challenge of superalignment, where the model reaches a regime in which humans can no longer provide reliable feedback. In such a regime, distilling a reward model from human feedback becomes infeasible. Therefore, specific results in [6], [7], and [8] are not self-improving approaches, as they rely on external reward models. That said, those loss functions can still be incorporated into self-rewarding training, as we demonstrated with DPO in our paper. Whether using an external reward model is advantageous depends primarily on the quality of the specific model. Thus, there is no contradiction. We are not arguing that external reward models shouldn’t be used; some external models may perform better in certain settings. Consequently, there will be scenarios where using external rewards is the better choice.

---

### Meta-Review · Area_Chair_uAan · 2024-12-14

**Metareview:**

The paper proposes self improving with a meta judge.

Strengths:
Timely topic
Clear writing
Interesting approach and techniques, with benefits (e.g. improving generation and verification at once)

Weaknesses:
There were concerns regarding the compute necessary, and the compute of this vs baseline methods.
It is unclear whether results (such as about the quality of models as a judge and a meta judge) would replicate to other models and are general.
See more proposals for improvements in reviews.

Overall the paper has merit but had enough open issues to call for addressing them before publishing.

**Additional Comments On Reviewer Discussion:**

Some of the rebuttal claims were left unanswered by the reviewers, despite public and private urging. However overall it seems there is enough to change.

---

### Decision · Program_Chairs · 2025-01-22

Reject